# Microbial Electrochemical CO₂ Reduction and In-Situ Biogas Upgrading at Various pH Conditions

**Wenduo Lu [1], Yuening Song [1], Chuanqi Liu [1], He Dong [1], Haoyong Li [1], Yinhui Huang [1], Zhao Liang [1], Haiyu Xu [2], Hongbin Wu [2], Pengsong Li [1], Dezhi Sun [1], Kangning Xu [1] and Yan Dang [1,*]**

[1]  Beijing Key Laboratory for Source Control Technology of Water Pollution, Engineering Research Center for Water Pollution Source Control and Eco-Remediation, College of Environmental Science and Engineering, Beijing Forestry University, Beijing 100083, China; luwenduo@yeah.net (W.L.)
[2]  Xinneng Qinglin (Beijing) Technology Co., Ltd., Beijing 100083, China
\*  Correspondence: yandang@bjfu.edu.cn; Tel.: +86-131-4123-6687

**Abstract:** Microbial electrochemical CO₂ reduction and in-situ biogas upgrading can effectively reduce the CO₂ content in biogas produced during anaerobic digestion, thereby reducing CO₂ emissions and achieving carbon reduction. pH is an important indicator in this process as it can significantly change the solubility and forms of CO₂ in the aquatic phase. This study comprehensively evaluated the optimal pH value from the perspectives of methane upgrading performance and electron utilization efficiency and observed and analyzed the morphology of the biofilm on the electrode surface and the microbial community in the cathodic region under optimal conditions. The results showed that the optimal pH was 6.5; methane content reached ~88.3% in the biogas; methane production reached a maximum of $22.1 \pm 0.1$ mmol·d$^{-1}$, with an increase in methane production compared to the control group reaching a maximum of 1.7 mmol·d$^{-1}$; and CO₂ conversion rate reached ~22.9%. A dense biofilm with a thickness of 51.3 μm formed on the electrode surface, with *Methanobacterium* being the dominant genus, with a high relative abundance of 69.3%, and *Geobacter* had a relative abundance of 20.1%. The above findings have important guiding significance for the practical application of methane upgrading.

**Keywords:** biogas upgrading; bioelectrochemical; methanogenesis; CO₂ bioreduction; anaerobic fermentation

## 1. Introduction

According to the report by the International Energy Agency (IEA) [1], global CO₂ emissions related to energy increased by 0.9% in 2022, reaching 36.8 gigatons (Gt), and the management of CO₂ emissions has become a crucial global issue. The anaerobic digestion of organic wastewater is an important method of wastewater resource utilization, which can convert organic matter into biogas. Biogas typically contains 50–70% CH₄ and 30–50% CO₂, traces of H₂, H₂S, N₂, NH₃ [2], as well as contaminants such as siloxanes [3] and VOCs [4]. The large presence of CO₂ severely affects the calorific value and subsequent use of biogas. So, it is critical to seek for economically green and efficient CO₂ conversion technologies, which will also be the development trend in energy gas treatment in Europe in subsequent decades [2]. The main technologies for CO₂ conversion include biotransformation [5], catalytic hydrogenation [6], photochemical or photoelectrochemical reduction [7], and electrochemical reduction [8,9]. Microbial electrochemistry is an emerging electrochemical reduction technology. At present, the main products of CO₂ reduction using the microbial electrochemical method include methane [10], alcohol (methanol, ethanol) [11], organic acids (such as formic acid, acetic acid, propionic acid, butyric acid) [12] and bioplastics (polyhydroxybutyric acid, PHB) [13]. Among them, the technology of separating or converting CO₂ from biogas to increase the concentration of CH₄ in biogas and generate biomethane [14,15] is called biogas upgrading.

Biogas upgrading technologies can be classified into in-situ upgrading and ex-situ upgrading. Ex-situ technologies for biogas upgrading include water scrubbing, pressure swing adsorption, physical scrubbing, chemical absorption, cryogenic separation, membrane separation, and biological upgrading technologies [16]. In comparison with ex-situ biogas upgrading technologies, in-situ biogas upgrading technologies present more economic advantages. Xu et al. [17] found that under an external cathodic potential of −0.7 V (vs. SHE), both intermittent-flow and continuous-flow in-situ biogas upgrading systems showed better $CO_2$ removal efficiency than the ex-situ one. In-situ technologies include $H_2$ addition [18], high-pressure anaerobic digestion (HPAD) [19], additives [20] and microbial electrochemistry [21]. Among them, microbial electrochemical biogas upgrading technology has the advantages of low cost, no pollution, good stability, and high product selectivity, and has attracted widespread attention in the fields of environment and energy [22]. However, its low upgrading efficiency and high energy consumption have also limited its widespread application.

In the anodic zone of microbial electrolysis cells (MECs), electrons and $H^+$ are produced, with electrons flowing through the external circuit to the microbial cathode and $H^+$ migrating to the cathodic zone through the cation exchange membrane. Under the microbial catalysis on the surface of the microbial cathode, $CO_2$ is directly reduced to $CH_4$ by the electron and $H^+$ via the direct electron transfer (DET) pathway, which requires less energy and is more efficient than the indirect electron transfer (IET) pathway mediated by a medium [23]. Some operational parameters have been found to affect the IET process. Lin et al. [24] studied four different cathode materials and found that the methane production rate of the reactor was highest (113.45 L/kg TS), and the phenol degradation rate was highest (52.3%) when carbon paper (CP) was used as the cathode. Carbon-based cathode materials are advantageous for the enrichment of microbes and the DIET process, while metal cathode materials can promote the transfer of $H_2$. Pelaz et al. [25] found that when the temperature decreased from 30 °C to 15 °C, the methane concentration decreased significantly due to the decrease in methane production activity. Mohanakrishna et al. [26] found that the concentration of bicarbonate can affect the rate of acetate production. The maximum acetate production rate was 142.2 $mg \cdot dm^{-3} \cdot day^{-1}$ when the bicarbonate concentration was 2.5 and 4 $g \cdot dm^{-3}$. pH has been shown to have a significant impact on MEC performance. Gao et al. [27] found that in the IET biogas upgrading process mediated by hydrogen, the accumulation of endogenous alkalinity could synergistically improve the biogas upgrading effect of the electrochemical methane production process. Currently, most research on the effect of pH on biogas upgrading efficiency focuses on the IET pathway, while research on the more efficient DET pathway is limited.

To explore the pH effects on the biogas upgrading performance in the bioelectrochemical system, a two-chamber microbial electrochemical reactor was constructed as an in-situ biogas upgrading anaerobic digestion system in this study. Firstly, the in-situ biogas upgrading efficiency of the direct electron transfer pathway and the reactor performance at different pH levels was comprehensively evaluated. Then, the electrochemical tests were carried out to demonstrate the electron utilization efficiency. After the optimal operating pH parameters were obtained, the morphology of the biofilm on the electrode surface and the microbial community structure in the cathodic zone were observed and analyzed to reveal its microbial mechanisms. The findings of this study provide theoretical and technical support for the practical application of biogas upgrading.

## 2. Materials and Methods

### 2.1. Reactor Setup and Operation

An improved two-chamber microbial electrochemical reactor (MEC) with a three-electrode system was constructed as an anaerobic treatment system (Figure 1). A cation exchange membrane (CEM; CMI-7000) (Membrane International Inc., NJ 07456) was used to separate the MEC reactor into an anodic chamber and a cathodic chamber. The anodic chamber was a cylindrical shape enclosed by the cation exchange membrane

($\phi$ 30 mm × 130 mm), while the cathodic chamber ($\phi$ 100 mm × 165 mm) had an effective working volume of approximately 1 L. Compared to traditional reactors, the improved MEC reactor had a larger cathodic chamber volume, which was more conducive to methane production. The cathode served as the working electrode and was made of carbon felt (120 mm × 50 mm × 3 mm). Before use, it was soaked in acetone for 24 h, followed by ethanol for 12 h, and then washed three times with deionized water under ultrasound before being dried for use, with the aim of removing excess impurities and adhesive materials from the surface of the carbon felt. The anode was made of a graphite rod ($\phi$ 6 mm × 140 mm), and the reference electrode was an Ag/AgCl electrode (+199 mV vs. SHE).

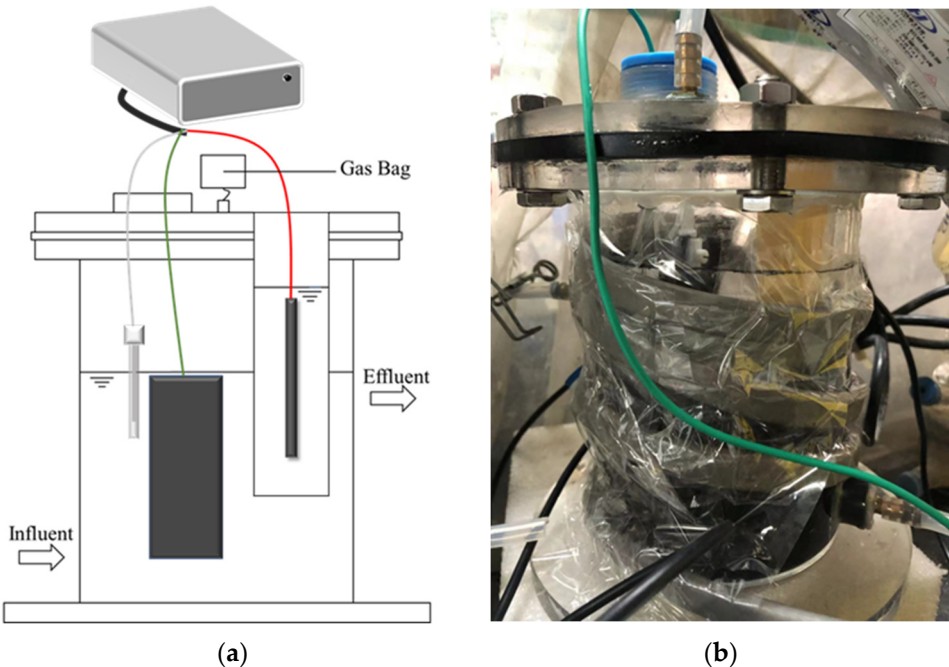

| **(a)** | **(b)** |

**Figure 1.** (**a**) Schematic diagram; (**b**) Photograph of the MEC reactors.

To start, 300 mL of anaerobic granular sludge from an actual anaerobic digestion process at a sewage treatment plant in Beijing was mechanically crushed and inoculated into the cathodic chamber, while the anodic chamber was left as a non-biological zone. Continuous-flow operation was employed. A peristaltic pump was used for inflow and outflow, with a flow rate of 0.55 L·d$^{-1}$ maintained for both inflow and outflow, and we used a hydraulic retention time of 1.8 d. The temperature was controlled at 37 ± 1 °C. The main components of the simulated wastewater were as follows (dm$^{-3}$): 2.0 mL methanol (to achieve an influent COD concentration of ~2400 mg·L$^{-1}$), 1.0 g NaHCO$_3$, 0.11 g KH$_2$PO$_4$, 0.11 g K$_2$HPO$_4$, 0.05 g Na$_2$SO$_4$, 0.05 g CaCl$_2$·2H$_2$O, 0.10 g MgCl$_2$·6H$_2$O, 5 mg NiCl$_2$·6H$_2$O, 6 mg CoCl$_2$·6H$_2$O, 2.5 mL DL Mineral solution, and 2.5 mL DL Vitamin solution. To balance the osmotic pressure, the anodic chamber solution was simulated wastewater without the addition of methanol or DL Vitamin.

An electrochemical workstation was used to precisely control the cathodic potential. In the experimental group, a potential of −400 mV (vs. SHE) was applied to the MEC reactor, while no potential was applied to the control group. The two sets of MEC reactors were operated at different pH levels, with the influent being adjusted to the corresponding pH for each reactor. The pH inside the reactors was monitored and adjusted daily. The pH range was selected between 6 and 8. Methanogens exhibit the highest activity within the pH range of 6.5 to 8.2, with the optimal pH being 7.0 [28]. When the pH is below 6.0, most methanogens will die; when the pH is higher than 8.0, a large amount of alkaline substances needs to be added, which is costly and not suitable for subsequent engineering applications. Additionally, physical adsorption is the main method used for CO$_2$ removal

at high pH, and $CO_2$ exists in an ion state in the liquid phase, which is not suitable for studying microbiological electrochemical biogas upgrading.

### 2.2. Analytical and Testing Methods

Gas produced during the reactor operation was collected using gas bags, and the composition and content of the gas were determined using a gas chromatograph (7890A, Agilent Technologies). The gas volume was measured using a syringe. A cyclic voltammetry (CV) test was performed using an electrochemical workstation (CS3104, Wuhan Corrtest Instruments Corp., Ltd.) in a three-electrode system, with a scanning range of $-900$ to 300 mV (vs. Ag/AgCl), a scanning rate of $5\ mV\cdot s^{-1}$, and a scanning frequency of 20 Hz. The total inorganic carbon (TIC) content in the aqueous solution was determined using a TOC/TN analyzer (TOC-L CSH, Shimadzu Corp., Japan). Scanning electron microscopy (SEM) was used to observe the morphology of the electrode surface microorganisms at magnifications of 5000 and 10,000 using a field emission scanning electron microscope (S-4800, Hitachi Ltd., Japan). The elemental composition of the biofilm was determined using an energy-dispersive X-ray spectroscopy (EDS) analyzer. The LIVE/DEAD™ BacLight™ Bacterial Viability Kit (L7012, Thermo Fisher Scientific) was used to stain the electrode surface, and laser confocal scanning microscopy (SP8, Leica Microsystems GmbH, Germany) was used to observe cell activity and biofilm thickness on the electrode surface. The excitation and emission wavelengths of SYTO™ 9 dye were 480/500 nm, while the excitation and emission wavelengths of propidium iodide (PI) were 535/617 nm.

The $CO_2$ conversion rate ($k_{conversion}$) was calculated according to the equation as follows (Equation (1)):

$$k_{conversion} = \frac{M_{conversion}}{M_{methanol}} \times 100\% = \frac{M_{CH_4} - 3M_{CO_2}}{M_{CH_4} + 3M_{CO_2}} \times 100\% \tag{1}$$

The equation provided below was used to calculate the corresponding Coulombic efficiency ($\eta_{CE}$) (Equation (2)).

$$\eta_{CE} = \frac{8\cdot(M_{MEC} - M_{Control})}{\int_0^t Idt/F} \times 100\% \tag{2}$$

In the equation, $I$ represents the current (mA) between the cathode and anode in the MEC reactor, $t$ represents time (s), and $F$ is the Faraday constant (96,485 $mC\cdot mmol^{-1}\ e^{-}$).

### 2.3. Microbial Analysis Methods

DNA extraction was performed on suspended sludge and electrode-attached biofilm from two reactors. The DNA extraction and purification process followed the instructions of the DNA rapid extraction kit from Beijing Edleader Biotechnology Co., Ltd. The extracted and purified DNA samples were sent to Majorbio Bio-Pharm Technology Co., Ltd. (Shanghai, China) for second-generation high-throughput sequencing. PCR amplification was performed using the primers listed in Table S1. High-throughput sequencing was performed on the Illumina Hiseq 2000 platform (Illumina, San Diego, CA, USA). Sequences with a similarity of more than 97% were merged into one operational taxonomic unit (OTU). Based on OTU clustering, community structure and composition were analyzed at various taxonomic levels.

### 2.4. Data Processing and Statistical Analysis

All experimental results are presented in the form of mean $\pm$ standard deviation. The t-test was performed using SPSS 26.0 (IBM, Armonk, NY, USA) to analyze the significant differences between the experimental group and the control group.

## 3. Results

### 3.1. Effect of pH on the Efficiency of Biogas Upgrading

Considering the methanogens' survival and optimal activities, as well as the $CO_2$ chemical adsorption properties under alkaline condition, the pH was chosen as 6.0~8.0 in this study. The variation of $CH_4$ concentration in biogas is shown in Figure 2a,b. At a pH of 6.0, 6.5, 7.0, and 7.5, the $CH_4$ concentration was significantly increased ($p < 0.01$) compared to the control group. At a pH of 6.5 and 7.0, the $CH_4$ concentration in both the experimental and control groups stabilized at around 88.3% and 85.2%, respectively, indicating a significant improvement in biogas upgrading, with an increase of 3.1% in $CH_4$ concentration. However, at a pH of 7.5, the $CH_4$ concentration in the control group increased to around 86.9%, while that in the experimental group remained at around 88.4%. When the pH increased to 8.0, the $CH_4$ concentration in both reactors increased significantly ($p < 0.01$), with the $CH_4$ concentration in the experimental group reaching 91.0%, as there was no significant difference compared to the control group. When the pH decreased to 6.0, the $CH_4$ concentration in the experimental group decreased to 84.9%. The increase in $CH_4$ concentration was significantly lower at a pH of 6.0, 7.5, and 8.0 than at a pH of 6.5 and 7.0 ($p < 0.01$). The microbial electrochemical biogas upgrading efficiency showed an increasing and then decreasing trend as pH increased from 6.0 to 8.0, with the optimal performance achieved under the neutral to slightly acidic conditions of pH 6.5 and 7.0.

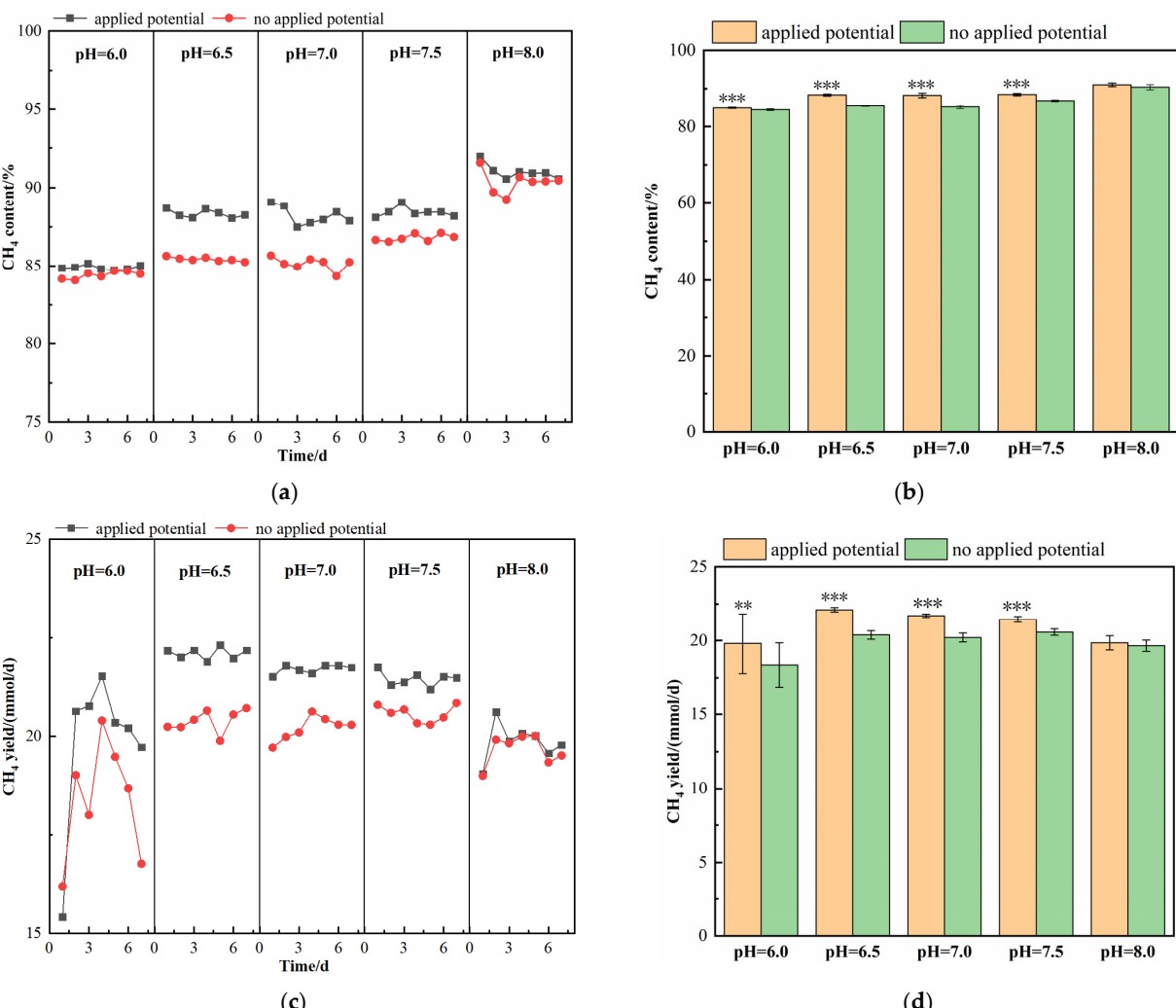

**Figure 2.** (**a**,**b**) $CH_4$ content; (**c**,**d**) $CH_4$ yield from the two MEC reactors. ** and *** represent the statistical significance of $p < 0.01$ and $p < 0.001$, respectively.

The variation of $CH_4$ production in biogas is shown in Figure 2c,d. At a pH of 6.0, 6.5, 7.0, and 7.5, the $CH_4$ production was significantly increased ($p < 0.01$) compared to the control group. At a pH of 6.5, the $CH_4$ production in the experimental group reached the maximum value of $22.1 \pm 0.1$ mmol·d$^{-1}$, and the increase in $CH_4$ production compared to the control group also reached the maximum value of 1.7 mmol·d$^{-1}$. When the pH increased to 7.0 and 7.5, the $CH_4$ production in the experimental group slightly decreased to $21.7 \pm 0.1$ mmol·d$^{-1}$ and $21.5 \pm 0.2$ mmol·d$^{-1}$, respectively. When the pH further increased to 8.0, the $CH_4$ production in the experimental group decreased to $19.9 \pm 0.5$ mmol·d$^{-1}$, and the $CH_4$ production in the control group also decreased to the same level. When the pH decreased to 6.0, the $CH_4$ production in the experimental group decreased to $19.8 \pm 2.0$ mmol·d$^{-1}$, but still increased by 1.5 mmol·d$^{-1}$ compared to the control group.

The variation in $CO_2$ concentration in biogas is shown in Figure 3a. At a pH of 6.0, 6.5, 7.0, and 7.5, the $CO_2$ concentration significantly decreased ($p < 0.01$). At a pH of 6.5 and 7.0, the $CO_2$ concentration in the experimental group decreased to approximately 11.6%, compared to approximately 14.7% in the control group. The $CO_2$ concentration showed a decrease of 3.0% relative to the control group. At pH 7.5, the $CO_2$ absorption increased due to the increase in pH, and the $CO_2$ concentration in the control group significantly decreased ($p < 0.01$) to approximately 13.0%, while the experimental group showed a decrease in $CO_2$ concentration to approximately 11.6%, a reduction of 1.4% compared to the control group.

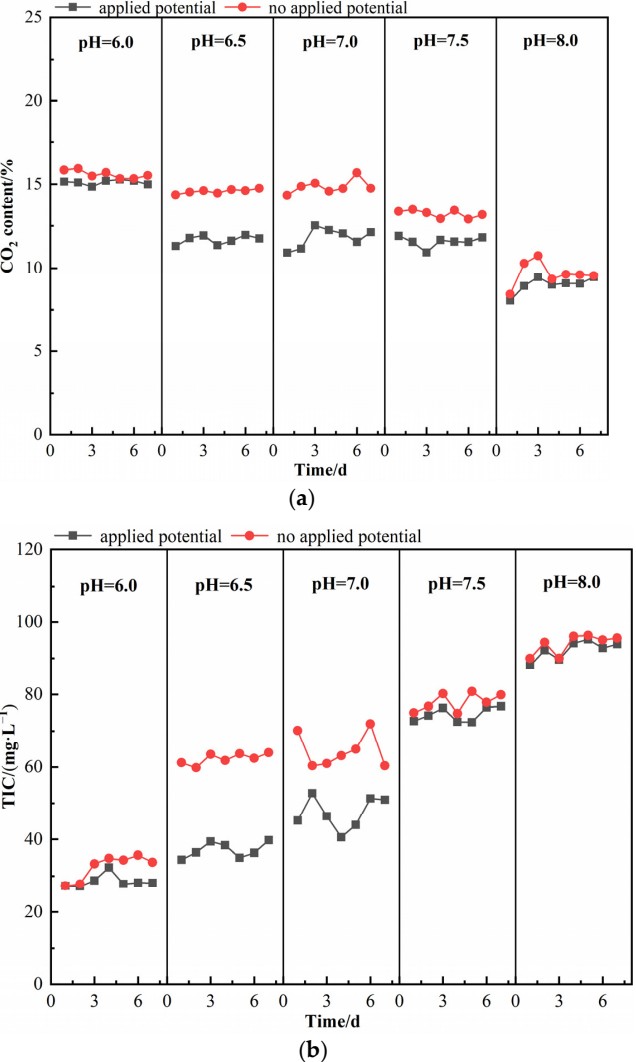

**Figure 3.** (**a**) $CO_2$ content; (**b**) TIC production from the two MEC reactors.

Throughout the operation of the reactor, the effluent TIC concentration remained higher than the influent TIC concentration. The TIC production was calculated by subtracting the influent TIC concentration from the effluent TIC concentration, as shown in Figure 3b. At a pH of 6.0, 6.5, 7.0, and 7.5, the TIC production significantly decreased compared to the control group ($p < 0.05$). At pH 6.5, the reduction in TIC concentration was the highest in the experimental group, reaching $25.2 \pm 2.0$ mg·L$^{-1}$, significantly higher than that at pH 7.0 ($17.3 \pm 6.8$ mg·L$^{-1}$).

As shown in Figure 4a, at a pH of 6.5 and 7.0, when the cathode potential was below $-0.6$ V, the electrode surface activity was completely stimulated, and the electron transfer rate on the cathode surface increased, resulting in a significant increase in current absolute value. This indicates that the electrochemical activity at a pH of 6.5 and 7.0 is significantly better than that at a pH of 6.0, 7.5, and 8.0, with slightly better electrochemical activity at pH 6.5 than at pH 7.0. The current changes at different pH levels are shown in Figure 4b. As shown in the figure, the absolute value of the current is ~12 mA at a pH of 6.5 and 7.0, while it decreases to 6 mA at pH 7.5 and continues to decrease to ~5 mA and ~3 mA at pH 6.0 and 8.0, respectively. At a pH of 6.5, 7.0, and 7.5, the Coulombic efficiency was greater than 100%. When pH was 6.5, the Coulombic efficiency was the highest, reaching $132.0 \pm 20.4\%$.

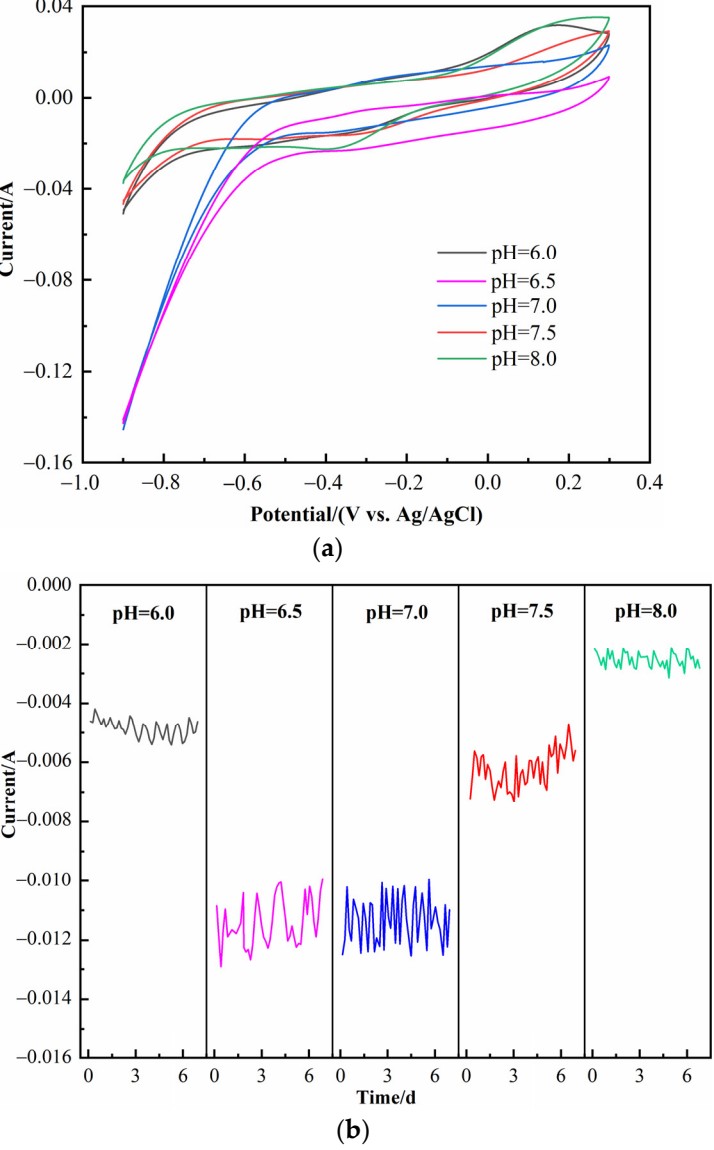

**Figure 4.** (**a**) Cyclic voltammetry curve; (**b**) Current generation.

### 3.2. Microbial Community Structure and Morphology of Cathode under Optimal Operating Conditions

#### 3.2.1. Biofilm Morphology of Cathode

Under the optimal operating conditions, the electrode surface was analyzed by scanning electron microscopy (SEM) after long-term reactor operation. As shown in Figure 5, there was almost no biofouling on the surface of the carbon fiber electrode without an applied potential, while a dense biofilm was formed on the electrode surface with an applied potential. Rod-shaped microorganisms were observed on the electrode surface, and further high-throughput analysis was conducted to determine the microbial community structure of the biofilm. Energy-dispersive X-ray spectroscopy (EDS) analysis indicated that Ca and Mg were the main inorganic elements accumulated on the electrode surface with an applied potential.

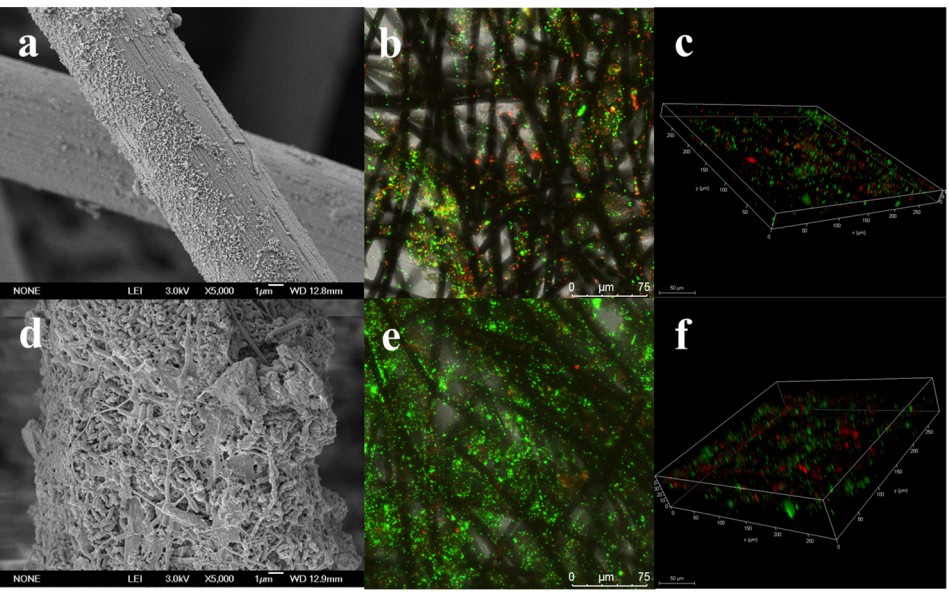

**Figure 5.** Biofilm morphology of cathode: SEM photographs of (**a**) no applied potential biocathode and (**d**) applied potential biocathode. CLSM of (**b**,**c**) no applied potential biocathode and (**e**,**f**) applied potential biocathode.

To further evaluate the film formation on the cathode surface, the electrode surface was observed using laser confocal microscopy after staining, as shown in Figure 5. As shown in the figure, there was no significant difference in overall cell viability on the electrode surface between the two groups of reactors. However, after the applied potential was provided, the biofilm thickness on the electrode surface significantly increased from 22.1 μm to 51.3 μm, an increase of 133%.

#### 3.2.2. Microbial Community Structure of Cathode

After the long-term operation of the reactors under optimal conditions, community analysis was conducted on the archaea and bacteria in the suspended sludge and on the electrode surface of the reactors with and without an applied potential. A diversity analysis of the communities showed that the Chao 1 index and Shannon index of the archaea and bacteria on the experimental group electrode surface samples were significantly lower than those in the suspended sludge and control group reactors, indicating lower microbial richness and diversity on the electrode surface in the experimental group with an applied potential. This suggests that the applied potential makes the electrode surface community more uniform and more functional for specific reactions.

According to Figure 6a, the dominant archaeal genera on the experimental group electrode surface were *Methanobacterium* and *Methanomethylovorans*, with relative abun-

dances of 69.3% and 19.7%, respectively. The relative abundance of *Methanobacterium* was significantly higher on the electrode surface of the experimental group than in the suspended sludge and control group electrode surface samples, indicating significant enrichment on the electrode surface under an applied potential. The relative abundance of *Methanomethylovorans* was similar in the experimental group suspended sludge (21.0%), significantly higher than in the control group suspended sludge (14.6%) and electrode surface (4.9%). *Methanomassiliicoccus* was more abundant in the control group suspended sludge (29.3%) and electrode surface (34.9%) than in the experimental group suspended sludge (16.1%) and electrode surface (5.1%).

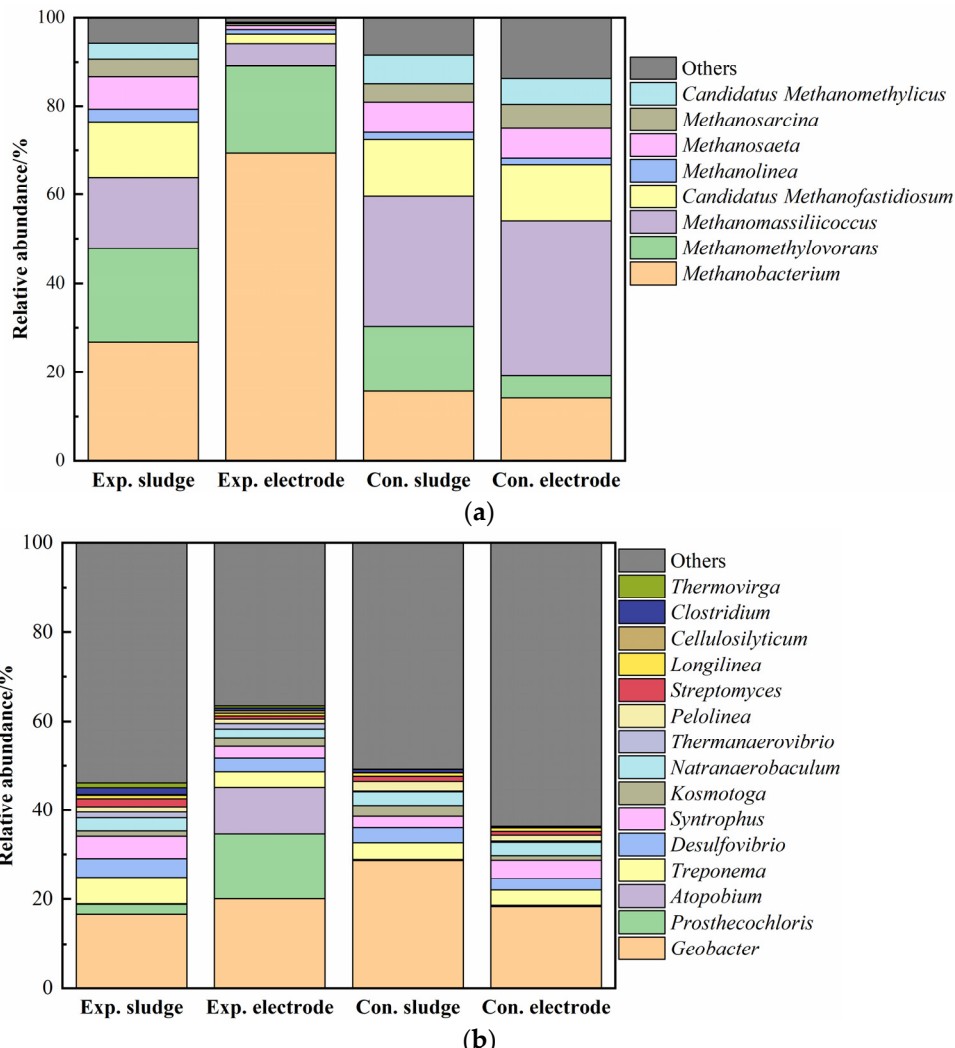

**Figure 6.** Proportion of major taxonomic groups of (**a**) archaea and (**b**) bacteria at genus level. Sequences belonging to taxa that make up less than 0.05% of all samples were classified under the "Others" category. "Exp." is an abbreviation for the experimental group, while "Con." is an abbreviation for the control group.

According to Figure 6b, the dominant bacterial genus on both suspended sludge and electrode surfaces of the two reactors was *Geobacter*, with the relative abundance on the experimental group electrode surface (20.1%) being higher than that on the control group electrode surface (18.3%). In contrast to the suspended sludge and electrode surface samples of the two reactors without an applied potential, the electrode surface with an applied potential was also enriched with *Prosthecochloris*, with relative abundances of 18.0%.

## 4. Discussion

According to the results of $CH_4$ concentration and production, it can be seen that the experimental group had the highest $CH_4$ concentration and production when pH was 6.5, and the effect of upgrading biogas was evident. It is suggested that in the weakly acidic environment of the reactor, some of the $HCO_3^-$ was converted into dissolved $CO_2$, which made it easier for methanogens to reduce the molecular $CO_2$ to produce $CH_4$, thereby increasing the $CH_4$ production. When pH increased to 7.5, the absorption of the solution for $CO_2$ increased, resulting in more $CO_2$ dissolving in water, which caused an increase in $CH_4$ concentration in the control group, while the $CH_4$ concentration in the experimental group did not change significantly. It is speculated that the increase in pH led to a decrease in the reduction amount of $CO_2$ and the increase in $CH_4$ concentration caused by $CO_2$ dissolving led them to cancel each other out. When pH increased to 8.0, the $CH_4$ concentration in both groups of reactors increased significantly and there was no significant difference, indicating that the increase in $CH_4$ concentration at this time was mainly due to the increase in the alkalinity of the solution and the synergistic effect of the biocatalytic reduction process on the electrode surface. The $CH_4$ production of both groups of reactors dropped to the same level, indicating that the decrease in $CH_4$ production at this time was mainly due to the decrease in activity of methanogens under alkaline conditions. When pH decreased to 6.0, due to the decrease in the solubility of $CO_2$ in water and the inhibition of the reduction process on the electrode surface under slightly acidic conditions, the $CH_4$ concentration in the experimental group decreased. Although the $CH_4$ production in the experimental group decreased, it still increased significantly compared with the control group, indicating that although the activity of methanogens was inhibited due to acidic conditions, the ability of the experimental group to resist acid impact was stronger than the control group, and the external potential strengthened the degradation process of methanol.

The production amount of TIC can reflect the efficiency of $CO_2$ conversion. The production of TIC is influenced by both the dissolution of gaseous $CO_2$ and the consumption in the microbial electrochemical reduction process ($CO_2 + 8H^+ + 8e^- \rightarrow CH_4 + 2H_2O$). In this study, the pH value inside the reactor was controlled, and the difference in the solubility of gaseous $CO_2$ caused by alkalinity between the experimental group and the control group could be neglected. When pH was 6.5, the experimental group had a maximum decrease in TIC concentration compared to the control group, and it was significantly higher than the decrease at pH 7.0. The $CO_2$ conversion rate was calculated to be ~22.9% at pH 6.5, significantly higher than that at other pH levels ($p < 0.01$), indicating that methanogens can convert more $CO_2$ at pH 6.5. Izadi et al. [29] found that in the process of the microbial electrosynthesis of acetic acid, acetic acid-producing bacteria tended to use $CO_2$ over bicarbonate.

The mechanism of upgrading biogas under different pH levels can be further revealed by electrochemical indicators. In this study, only $H^+$ and $CO_2$ in water can serve as the final electron acceptor, and interspecies $H_2$ transfer ($CO_2 + 4H_2 \rightarrow CH_4 + 2H_2O$ $\Delta E = -0.614$ V vs. Ag/AgCl) and direct electron transfer ($CO_2 + 8H^+ + 8e^- \rightarrow CH_4 + 2H_2O$ $\Delta E = -0.444$ V vs. Ag/AgCl) may occur. Due to the existence of overpotential, the current generated at $-0.6$ V in CV is most likely the reaction of a direct reduction of $CO_2$ to $CH_4$. Compared to traditional interspecies $H_2$ transfer, DET has better metabolic utilization and higher energy efficiency. Based on these indices, it can be seen that the methane production efficiency deteriorates at pH 6.0, 7.5, and 8.0. The main reason for this is that at a pH of 7.5 and 8.0, the microbial activity decreases and the required amount of $H^+$ for $CO_2$ reduction decreases, resulting in a significant decrease in the absolute value of the current from ~12 mA to ~6 mA and ~3 mA, respectively. As a result, $CO_2$ obtains fewer electrons, and the reduction of $CO_2$ and the production of $CH_4$ decrease. Although there is sufficient $H^+$ in the system at pH 6.0, the activity of methanogens is reduced under acidic conditions, leading to the inhibition of the biocatalytic reduction process.

The morphology of the biofilm on the electrode surface in the cathodic region was observed and analyzed at the optimal pH. Based on the results of SEM and CLSM, it

was found that a large number of microorganisms were enriched on the surface of the electrode under applied potential, growing into a dense biofilm. This indicates that the applied potential provided energy to promote microbial growth on the electrode surface and allowed microorganisms to directly obtain electrons from the electrode surface to reduce $CO_2$ and produce $CH_4$. Previous studies have found a correlation between the biomass and the potential of the microbial cathode, and the improvement of cathode performance can be attributed to the higher biomass formed on the cathode [30]. According to EDS results, it was speculated that Ca and Mg were deposited on the electrode surface under applied potential mainly due to the consumption of $H^+$ on the cathode and the slower mass transfer rate of cation exchange membranes that limited the $H^+$ transport from the anode to the cathode. The cathode-adjacent environment was weakly alkaline, and $OH^-$ combined with Ca and Mg in water to form inorganic salt deposition on the cathode surface, leading to scaling. Inorganic substances can promote the tight connection between microorganisms and their metabolites, enhancing the strength of the biofilm [31], thereby making the reactor operation more stable.

*Methanobacterium* and *Geobacter* are enriched on the surface of the applied potential electrode. The genus *Methanobacterium* is a group of hydrogenotrophic methanogenic archaea. Previous studies have shown that *Methanobacterium* is the predominant group of methanogenic bacteria on the surface of the biocathode during electrochemical methano-genesis [32], capable of directly accepting electrons from the electrode to reduce $CO_2$ to $CH_4$ [21]. The species YSL within this genus has been shown to co-culture with *Geobacter sulfurreducens* through direct interspecies electron transfer (DIET) [33]. The significant enrichment of the *Methanobacterium* genus on the electrode surface in the experimental group indicates the potential of the applied potential system to reduce $CO_2$ via the DET pathway. *Methanomethylovorans* is a methanol-utilizing methanogen, belonging to the *Methanosarcinaceae* family, and is closely related to acetoclastic methanogens that can perform DIET with *Geobacter* [34]. *Methanomassiliicoccus* grows on methanol or methylamine with $H_2$ as the electron donor but does not reduce $CO_2$ to $CH_4$ [35]. *Geobacter* can transfer electrons generated from organic matter degradation to methanogens that can directly accept electrons via conductive pili or extracellular cytochromes. Previous studies have shown that *Prosthecochloris aestuarii* can directly acquire electrons from the electrode and also receive extracellular electrons from *Geobacter sulfurreducens* via DIET, a process closely related to the presence of $CO_2$ [36]. The enrichment of *Prosthecochloris* on the experimental group electrode surface suggests its potential involvement in electron transfer. Other bacterial genera also play a positive role in methane production by participating in electron transfer. For instance, *Syntrophus aciditrophicus* of the *Syntrophus* genus can produce conductive pili and grow via DIET [37]; the dominant *Kosmotoga* genus in the reactor with added $Fe_3O_4$ is related to the DIET process [38]; *Longilinea* is a bacterium with electroactivity [39]; and the enrichment of *Clostridium* can promote the abundance of *Geobacter* due to their cooperative relationship in the complex electroactive biofilm [40].

## 5. Conclusions

This study investigated the in-situ upgrading performance of direct electron transfer pathways under different pH conditions. The results showed that the optimal pH for methane upgrading was 6.5, with methane concentration reaching ~88.3%, methane production reaching a maximum of $22.1 \pm 0.1$ mmol·d$^{-1}$, and $CO_2$ conversion rate reaching ~22.9%. The morphology of the biofilm on the electrode surface and the microbial community structure in the cathodic region were observed and analyzed at the optimal pH. The thickness of the biofilm on the electrode surface was 51.3 μm, with *Methanobacterium* being the dominant genus, with a high relative abundance of 69.3%, and *Geobacter* had a relative abundance of 20.1%. The findings of this study are of great importance as they can be applied in real anaerobic digestion projects and provide an in-situ biogas upgrading method, which will significantly improve the methane content in the output biogas and enhance the biogas usability.

**Supplementary Materials:** The following supporting information can be downloaded at: https://www.mdpi.com/article/10.3390/fermentation9050444/s1, Figure S1: COD concentrations in the influent and effluent from the two reactors; Figure S2: archaeal community at the phylum level; Figure S3: bacterial community at the phylum level; Table S1: Primer sequences for high throughput sequencing.

**Author Contributions:** Conceptualization, W.L. and Y.D.; methodology, W.L. (experiment design) and C.L. (reactor design); validation, W.L., Y.S. (gas content test and replication) and C.L. (water characteristic test and replication); formal analysis, W.L.; investigation, W.L.; data curation, H.D. (electrochemical data), H.L. (microbial community data), Y.H. (gas compound data) and Z.L. (water characteristic data); writing—original draft preparation, W.L.; writing—review and editing, Y.D. (reactor performance analysis), K.X. (introduction and material and methods) and P.L. (microbes analysis); supervision, D.S. and Y.D.; funding acquisition, H.W. and H.X. (funding providers for this projects). All authors have read and agreed to the published version of the manuscript.

**Funding:** This research was funded by National Key Research and Development Program of China (No. 2021YFC3200602), the National Natural Science Foundation of China (No. 52270023), the horizontal scientific research project "Technology developments and engineering demonstration of efficient anaerobic digestion treatment for livestock manure and kitchen waste" (No. 2022-HXKF-058), and the Beijing Municipal Education Commission for their financial support through Innovative Transdisciplinary Program "Ecological Restoration Engineering" (No. GJJXK210102).

**Institutional Review Board Statement:** Not applicable.

**Informed Consent Statement:** Not applicable.

**Data Availability Statement:** Data is unavailable due to privacy.

**Conflicts of Interest:** The authors declare no conflict of interest.

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
