# Peer review of "Microbial Electrochemical CO2 Reduction and In-Situ Biogas Upgrading at Various pH Conditions"

_fermentation, doi:10.3390/fermentation9050444_

Round 1
Reviewer 1 Report
I suggest expanding the discussion and increasing the number of References.
29: "[1]" - I suggest not using superscript.
58: "·L" - I suggest writing dm3.
71 - At the end of the Introduction chapter, I suggest indicating the aim of the work.
I have no objections.
Author Response
Reviewers' comments:
Reviewer #1:
I suggest expanding the discussion and increasing the number of References.
Re: As suggested by the reviewer, we have expanded the discussion and added relevant references in the manuscript (now Lines 613-707).
29: "[1]" - I suggest not using superscript.
Re: As suggested by the reviewer, we have adjusted the superscript.
58: "·L" - I suggest writing dm3.
Re: As suggested by the reviewer, we have adjusted the “L” to “dm3”.
71 - At the end of the Introduction chapter, I suggest indicating the aim of the work.
Re: As suggested by the reviewer, we have added the aim of our work in the end of the Introduction chapter.
Reviewer 2 Report
Authors presented for evaluation manuscript entitled "Microbial Electrochemical CO2 Reduction and in-situ Biogas Upgrading at Various pH conditions" the subject is actual and in my opinion should be presented for scientific community. However, some revisions are needed:Add in the Introduction the topic of biogas and its upgrading to biomethane quality it is the trend in EU. Consider recent article https://doi.org/10.1016/j.energy.2022.125319
1. In the Intro section add the novelty aspect of your research and highlight the specific your research in context to existing work. and add the information on sicontaminats in biogas (incl. siloxanes: https://doi.org/10.3390/molecules26071953 and VOCs) 2. Results: Add the statistical data (!)
3. Add the error bars on the charts
4. Add the practical meaning of you research
5. Table 1. Is brief, add more details or remove
6. Extend the discussion (more citation is needed) with paragraph 3.2
The language has to be polished.
Author Response
Reviewer #2:
Authors presented for evaluation manuscript entitled "Microbial Electrochemical CO2 Reduction and in-situ Biogas Upgrading at Various pH conditions" the subject is actual and in my opinion should be presented for scientific community. However, some revisions are needed:
Add in the Introduction the topic of biogas and its upgrading to biomethane quality it is the trend in EU. Consider recent article https://doi.org/10.1016/j.energy.2022.125319
Re: As suggested by the reviewer, we have added the relevant information and cited this reference.
- In the Intro section add the novelty aspect of your research and highlight the specific your research in context to existing work. and add the information on sicontaminats in biogas (incl. siloxanes: https://doi.org/10.3390/molecules26071953 and VOCs)
Re: As suggested by the reviewer, we have highlighted the novelty of our work and added the relevant information, as well as cited this reference.
- Results: Add the statistical data (!)
Re: As suggested by the reviewer, we have added section 2.4 Data processing and statistical analysis. And we also added the statistical significance in the Fig.2(b) and (d) to perform statistical analysis.
- Add the error bars on the charts
Re: As suggested by the reviewer, we have added the error bars on the charts (Fig.2(b) and (d)).
- Add the practical meaning of you research
Re: As suggested by the reviewer, we have added the practical meaning of our research (now Lines 674-677).
- Table 1. Is brief, add more details or remove
Re: As suggested by the reviewer, we have removed Table 1 from the manuscript to the supplementary material.
- Extend the discussion (more citation is needed) with paragraph 3.2
Re: As suggested by the reviewer,we have extended the discussion with Section 3.2 and included it in the Discussion section and added more references.
Reviewer 3 Report
Dear authors, after carefully reading your article, I have a few comments, the correction of which will significantly improve the perception of the article.
1. In my opinion, the article has a very large number of authors (13). It is necessary to describe in more detail the contribution of each author to obtaining the results.
2. It is desirable to update data on CO2 emissions for 2022 (Introduction section).
3. In the introduction section, it is necessary to highlight the subsection Literature review. This subsection should contain more sources used.
4. At the end of the introduction section, you need to insert information about the further structure of the article.
5. In subsection 2.1, it is necessary to provide a diagram or a photograph of the main laboratory equipment used.
6. Authors should explain why the pH range of 6-8 was chosen.
7. Fig. 3 in black and white, and all others in color. It is necessary that there be uniformity in the article.
8. It is necessary to separate the sections Results and Discussion.
9. Mathematical dependencies and their statistical evaluation can be added to the article.
Author Response
Reviewer #3:
Dear authors, after carefully reading your article, I have a few comments, the correction of which will significantly improve the perception of the article.
- In my opinion, the article has a very large number of authors (13). It is necessary to describe in more detail the contribution of each author to obtaining the results.
Re: As suggested by the reviewer, we have added more detail description of the contribution of all the authors in the section “Author Contributions” .
- It is desirable to update data on CO2 emissions for 2022 (Introduction section).
Re: As suggested by the reviewer, we have updated the data about CO2 emissions of 2022 (Line 30).
- In the introduction section, it is necessary to highlight the subsection Literature review. This subsection should contain more sources used.
Re: As suggested by the reviewer, we have significantly improved the literature review in the Introduction section and added 10 more references.
- At the end of the introduction section, you need to insert information about the further structure of the article.
Re: As suggested by the reviewer, we added the information about the further structure of the article at the end of the Introduction section (now Lines 101-107).
- In subsection 2.1, it is necessary to provide a diagram or a photograph of the main laboratory equipment used.
Re: As suggested by the reviewer, we have provided the schematic diagram and photograph of the MEC reactors (now Fig.1).
- Authors should explain why the pH range of 6-8 was chosen.
Re: We understand the reviewers’ concern. Actually, methanogens exhibit the highest activity within the pH range of 6.5 to 8.2, with the optimal pH being 7.0. When the pH is below 6.0, most methanogens cannot survive; when the pH is higher than 8.0, a large amount of alkaline substances need to be added, which is costly and not suitable for subsequent engineering applications. Additionally, chemical adsorption is the main method used for CO2 removal at high pH, and CO2 exists in an ion state in the liquid phase, which is not suitable for studying microbiological electrochemical biogas upgrading.
We have now included the above information in the manuscript (now Lines 223-225).
- Fig. 3 in black and white, and all others in color. It is necessary that there be uniformity in the article.
Re: As suggested by the reviewer, we have modified the color of Fig. 3 (Now Fig.4).
- It is necessary to separate the sections Results and Discussion.
Re: As suggested by the reviewer, we have separated the sections Results and Discussion.
- Mathematical dependencies and their statistical evaluation can be added to the article.
Re: As suggested by the reviewer, we have added section 2.4 Data processing and statistical analysis. And we also added the statistical significance in the Fig.2(b) and (d) to perform statistical analysis.
Round 2
Reviewer 2 Report
Dear Authors,
The revised version of your work in my opinion can be recommended for publication, therefore I look forward the final & published version of this ms.
Is ok, can be checked.
Reviewer 3 Report
Dear authors, you have made good work. The article in its present form is an excellent illustration of your study.